# STAT3 serine phosphorylation is required for TLR4 metabolic reprogramming and IL-1β expression

Jesse J. Balic [1,2,10], Hassan Albargy[1,2,10], Kevin Luu[1,2], Francis J. Kirby[1,2], W. Samantha N. Jayasekara[2,3], Finbar Mansell[1,2], Daniel J. Garama[2,3], Dominic De Nardo[4], Nikola Baschuk[1,2], Cynthia Louis[5,6], Fiachra Humphries[7], Katherine Fitzgerald [7], Eicke Latz [8,9], Daniel J. Gough [2,3✉] & Ashley Mansell [1,2✉]

Detection of microbial components such as lipopolysaccharide (LPS) by Toll-like receptor 4 (TLR4) on macrophages induces a robust pro-inflammatory response that is dependent on metabolic reprogramming. These innate metabolic changes have been compared to aerobic glycolysis in tumour cells. However, the mechanisms by which TLR4 activation leads to mitochondrial and glycolytic reprogramming are unknown. Here we show that TLR4 activation induces a signalling cascade recruiting TRAF6 and TBK-1, while TBK-1 phosphorylates STAT3 on S727. Using a genetically engineered mouse model incapable of undergoing STAT3 Ser727 phosphorylation, we show ex vivo and in vivo that STAT3 Ser727 phosphorylation is critical for LPS-induced glycolytic reprogramming, production of the central immune response metabolite succinate and inflammatory cytokine production in a model of LPS-induced inflammation. Our study identifies non-canonical STAT3 activation as the crucial signalling intermediary for TLR4-induced glycolysis, macrophage metabolic reprogramming and inflammation.

[1] Centre for Innate Immunity and Infectious Diseases, Hudson Institute of Medical Research, Clayton, VIC 3168, Australia. [2] Department of Molecular and Translational Sciences, Monash University, Clayton, VIC 3168, Australia. [3] Centre for Cancer Research, Hudson Institute of Medical Research, Clayton, VIC 3168, Australia. [4] Department of Anatomy and Developmental Biology, Monash Biomedicine Discovery Institute, Monash University, Clayton, VIC 3800, Australia. [5] Inflammation Division, The Walter and Eliza Hall Institute of Medical Research, 1G Royal Parade, Parkville, VIC 3052, Australia. [6] Medical Biology, University of Melbourne, Parkville, VIC, Australia. [7] Division of Infectious Diseases and Immunology, University of Massachusetts Medical School, Worcester, MA, USA. [8] Institute of Innate Immunity, University Hospital Bonn, University of Bonn, Bonn, Germany. [9] German Center for Neurodegenerative Diseases (DZNE), Bonn, Germany. [10] These authors contributed equally: Jesse J. Balic, Hassan Albargy. ✉email: daniel.gough@hudson.org.au; ashley.mansell@hudson.org.au

TLR4 activation induces metabolic changes in macrophages via mitochondrial reprogramming, which is required to meet the rapid increase in demand for biosynthetic precursors for lipids, proteins, nucleic acids and the increased energy demand of the inflammatory state[1–4]. LPS-induced transcription of inflammatory cytokines and chemokines is also dependent on metabolic reprogramming[2,5].

Activated macrophages become more glycolytic, increase reactive oxygen species (ROS) production, and accumulate the tricarboxylic acid (TCA) cycle metabolite succinate redirecting it away from oxidative phosphorylation via the electron transfer chain (ETC). Recent studies have identified that the TLR downstream kinases TBK-1 and IKKε play a role in inducing glycolysis in immune cells[6,7] via phosphorylation of Akt[6], although how these signals converge on, and orchestrate mitochondrial metabolic function remain unclear.

Metabolic signalling in response to elevated succinate concentration leads to the stabilisation of hypoxia inducible factor-1α (HIF-1α), which positively regulates the prototypic inflammatory cytokine IL-1β and other HIF-1α-dependent genes involved in glycolysis[8]. However, the molecular mechanisms of how membrane-bound TLRs communicate this signal to the mitochondria to alter mitochondrial function are also unclear.

STAT3 is a critical signalling molecule activated by immune cytokines resulting in phosphorylation on Tyr705 and activation of its activity as a transcription factor[9]. In addition to this essential activity in the nucleus we, and others, have shown that a pool of Ser727 phosphorylated STAT3 translocates into the mitochondria where it affects mitochondrial metabolism and ROS generation[10,11,12].

In this study we interrogate the activation of STAT3 by TLRs as a mechanism to directly alter mitochondrial metabolism. STAT3 interacts directly with TLR signalling pathways via interaction with TRAF6 and TBK-1, resulting in non-canonical STAT3 Ser727 phosphorylation. STAT3 pSer727 is subsequently required for TLR-induced mitochondrial reprogramming and production of inflammatory cytokines. These studies identify a function for STAT3 in TLR-induced metabolic reprogramming and inflammation.

## Results and discussion

**TRAF6 interacts with STAT3 following TLR4 challenge.** Activation of TLRs, with the exception of TLR3, sequester MyD88 to the receptor complex, initiating recruitment of the serine kinases IRAK1 and IRAK4 into the Myddosome complex[7,13–16]. This complex subsequently interacts with TRAF6 to activate the canonical signalling pathway, resulting in the nuclear translocation of NF-κB and induction of the classic pro-inflammatory response[17]. Alternatively, TLR3 and TLR4 also engage the adaptor TRIF leading to the activation of the serine kinase TBK-1 and subsequent phosphorylation of IRF3 leading to nuclear translocation and induction of IFNβ expression.

TRAF6 represents a major point of bifurcation of TLR signalling between canonical and non-canonical induction of inflammation. Structural analysis of TRAF6-binding partners show a conserved binding motif consisting of Pro-X-Glu-X-X-(aromatic or acidic residue)[18], required for interaction with downstream signalling proteins including Mal/TIRAP, TRIF, TRAM and STAT1[19–22]. Given the homology between STAT proteins and the role for STAT3 in mitochondrial reprogramming, we identified highly conserved putative TRAF6-binding motifs within STAT3 (Fig. 1a) and confirmed that the interaction between endogenous TRAF6 and STAT3 occurs within 10 min of LPS-stimulation in macrophages (Fig. 1b). To ascertain the importance of the putative TRAF6 binding sites in STAT3 we

generated a series of mutant STAT3 constructs, which were transiently expressed in 293T cells and bound to recombinant GST-tagged TRAF6 ex vivo. This data revealed selectivity for Glu100, i.e. STAT3 E100A completely abolished the interaction with TRAF6, which was not observed for the other putative interaction motifs (Fig. 1c).

**TLRs induce STAT3 Ser, by not Tyr, phosphorylation.** Initially, we treated immortalised bone marrow-derived macrophages (iBMDMs) with IFNα as a well-known activator of STAT3. While we observed background Ser727 phosphorylated STAT3 in untreated cells, iBMDMs exhibit rapid increases in phosphorylation on both Ser727 and Tyr705 (Fig. 2a) as expected. In contrast TLR ligands are reported to initiate phosphorylation of STAT3 on Tyr705 only after prolonged stimulation more consistent with secondary and indirect activation. To resolve whether TLR activation induced rapid STAT3 phosphorylation, we challenged primary bone marrow-derived macrophages (BMDMs) with LPS and observed S727 phosphorylation within 20 min of challenge, whereas Tyr705 phosphorylation was not detected even after 120 min (Fig. 2b). To establish whether STAT3 Ser727 phosphorylation was induced by all TLRs we stimulated macrophages with Pam3Cys (TLR2), poly I:C (TLR3), Loxoribine (TLR7), CpG-DNA (TLR9) or IFNα as a positive control. All TLR agonists induced Ser727, but not Tyr705 phosphorylation of STAT3, whilst IFNα induced both pSer727 and pTyr705 as expected (Fig. 1c).

Stimulation of the TLR signalling pathway has at least two phases of response. The early response which is dependent upon the Myddosome complex components MyD88 and TRIF, leading to the activation of NF-κB and Interferon regulatory factor (IRF)-3 which drive the transcription of a suite of inflammatory genes including type I IFN[17]; and the secondary response to secreted type I IFN. To confirm that STAT3 Ser727 phosphorylation is due to TLR activation directly and not as a consequence of autocrine cytokine or type I IFN signalling, we established the phosphorylation of STAT3 on S727 in cells lacking the critical Myddosome factor MyD88 and TRIF or lacking the type I IFN receptor (IFNAR1). We show that LPS does not induce STAT3 phosphorylation on S727 in MyD88$^{-/-}$/TRIF$^{-/-}$ iBMDMs (Fig. 1d) but is unaffected in IFNAR1$^{-/-}$-deficient macrophages (Fig. 1e). Taken together these results demonstrate, for the first time, that STAT3 is directly recruited into the TLR signalling pathway via interaction with TRAF6, resulting in STAT3 Ser727, but not Tyr705 phosphorylation.

TLR-induced Ser727 phosphorylation of STAT3, in the absence of Tyr705 phosphorylation is consistent with the activation of the novel mitochondrial STAT3 activity that we have previously described in RAS transformed cancer cells[11]. While the majority of STAT3 is present in the cytosol, a pool of STAT3 translocates into the mitochondria, causing metabolic reprogramming, and altering ROS production, dependent on pSer727 but independent of Tyr705 phosphorylation[11]. We therefore, performed biochemical fractionation of LPS-treated macrophages (Fig. 1f) and demonstrate an enrichment of STAT3 pSer727 in mitochondrial fractions after 60 min LPS stimulation. Taken together, these results demonstrate that STAT3 undergoes rapid TLR-induced Ser727, but not Tyr705 phosphorylation and an accumulation in the mitochondrial fraction of macrophages.

**TBK-1 is required for TLR4-mediated STAT pSer727.** We next wished to identify the potential kinase responsible for TLR-induced STAT3 Ser727 phosphorylation. We have previously shown that the mitochondrial activity of STAT3 results in increased mitochondrial ROS (mtROS) production[23]. Therefore,

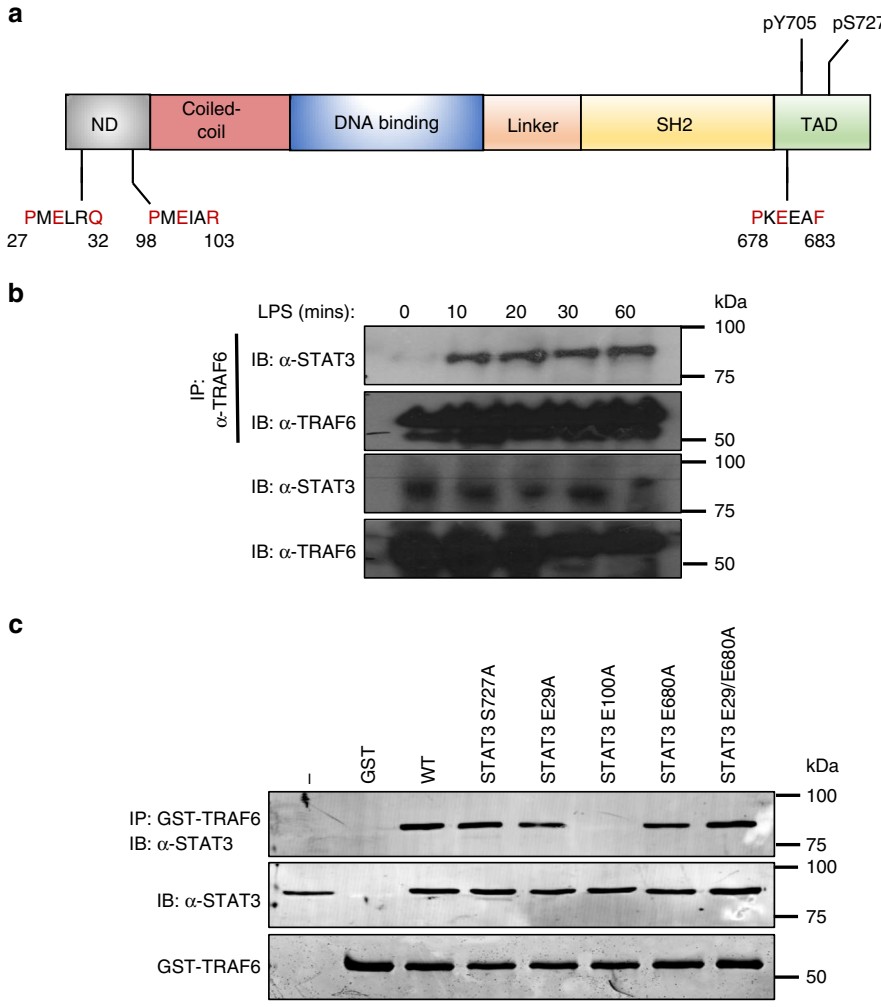

**Fig. 1 STAT3 directly interacts with TRAF6 after LPS stimulation. a** Schematic representation of the putative TRAF6-binding domains and phosphorylation sites of STAT3. ND: N-terminal domain: SH2: Src homology domain TAD: transactivation domain. **b** BMDMs ($1 \times 10^7$ cells) were seeded in 10 cm culture dishes and grown for 24 h prior to stimulation with LPS (1 μg/ml) for indicated times. TRAF6 was immunoprecipitated from cellular lysates and probed with anti-TRAF6, or STAT3 antibody ($n = 3$) **c** HEK293T cells ($1 \times 10^6$) were transfected with indicated plasmid vectors for 24 h. Cells were lysed and probed with recombinant GST-TRAF6 to immunoprecipitated interacting proteins, separated by SDS-PAGE and probed with anti-FLAG antibody ($n = 3$).

to identify potential kinases responsible for LPS-induced STAT3 Ser727 phosphorylation we used mtROS concentration as a functional readout of activity and screened a library of 355 kinase inhibitors (Supplementary Dataset 1). Macrophages were pretreated with inhibitors for 30 min prior to LPS challenge and mtROS production monitored every hour for 4 h (Supplementary Fig. S1) using the mitochondrial superoxide probe MitoSOX. As a positive control we included the ROS scavenger N-Acetyl-L-cysteine in all screens. This screening approach identified several classes of inhibitors capable of reducing LPS-induced mtROS production (Fig. 2a); including PI3K, mTOR and Inhibitor of IκB kinase (IKK) inhibitors. Interestingly, we also identified an inhibitor of tank binding kinase-1 (TBK-1) as a suppressor of LPS-induced mtROS production. Importantly, IRAK1 and IRAK4 are serine kinases that interact with TRAF6[18] and the MyDDosome whose kinase activity is critical for TLR-signalling and NF-κB activation[13]. We found however, that a specific inhibitor of IRAK-1/4 activity had no effect on LPS-induced mtROS production, and therefore may not be responsible for STAT3 phosphorylation, acting as a specificity control for this screen.

This is consistent with the recent publication from Tan and Kagan[7] who showed that TRAF6 depleted macrophages are

defective for TBK-1 recruitment to the Myddosome and induction of TLR glycolysis. Analysis of the TBK-1 sequence revealed a putative TRAF6 binding motif (human amino acids 223–228) suggesting the potential interaction between these proteins. Indeed, immunoprecipitation of endogenous protein show that TBK-1 directly interacts with TRAF6 following LPS stimulation in a time-dependent and transient manner (Fig. 2b). This TBK-1-TRAF6 interaction was observed within 10 min of LPS stimulation and was no longer detectable after 120 min post-stimulation. Our previous data demonstrate interaction between TRAF6 and STAT3 and led us to propose the formation of a TBK-1, TRAF6, STAT3 complex to enable TBK-1 phosphorylation of STAT3 in response to LPS challenge. Consistent with this we show that STAT3 interacted with phosphorylated TBK-1, with kinetics that are slightly offset from that of the TRAF6-TBK-1 interaction (Fig. 2c). Importantly, the immunoprecipitated STAT3 was phosphorylated on Ser727.

Whilst interaction studies and our mtROS inhibitor screen suggest that TBK-1 is the kinase upstream of STAT3 S727 phosphorylation in response to LPS stimulation we wanted to formally test this. We therefore examined STAT3 S727 phosphorylation in TBK-1-deficient macrophages. As can be

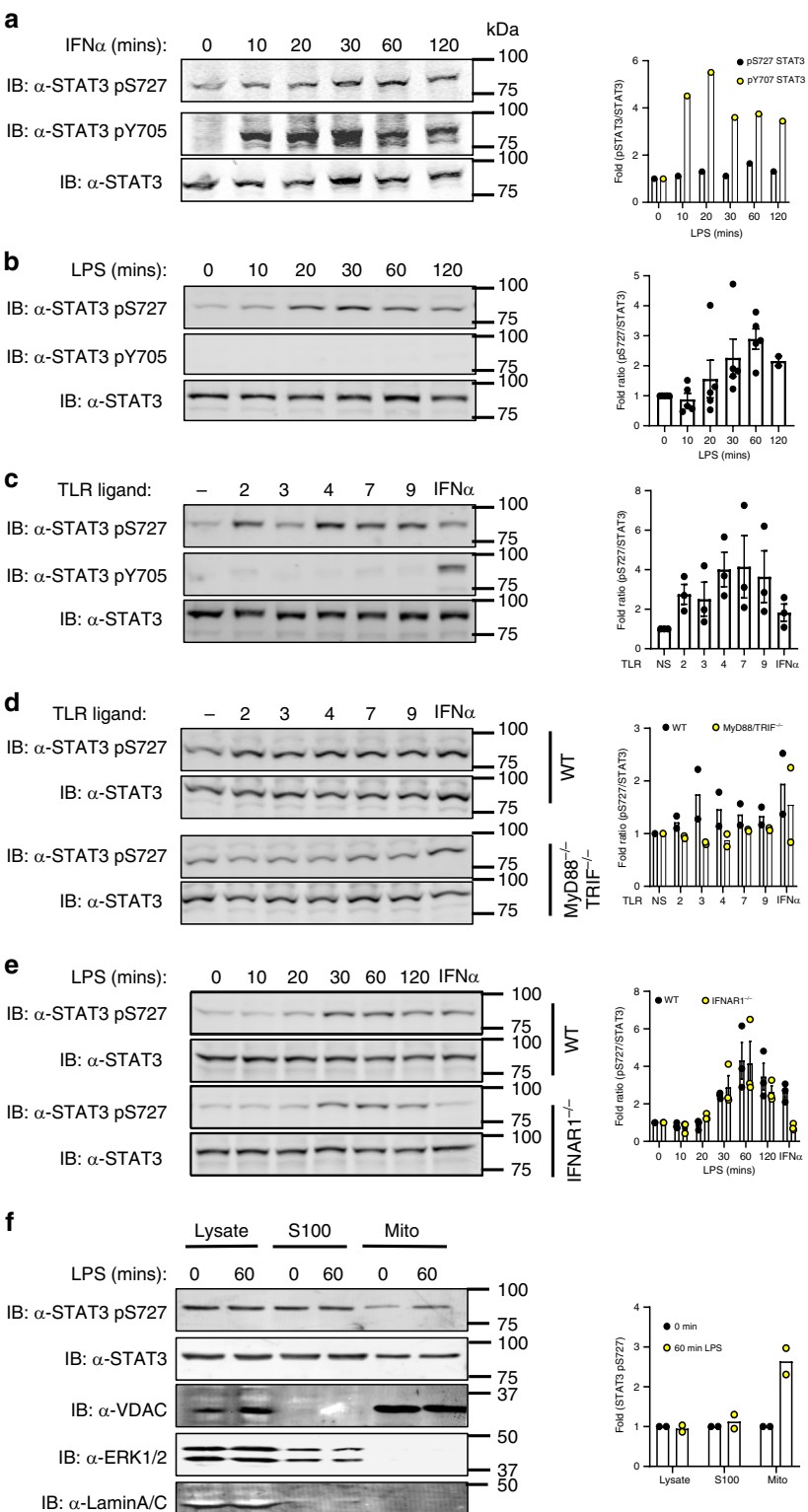

seen in Fig. 3d, we observed a consistent reduction in STAT3 S727 phosphorylation of ~50% when compared with WT cells, suggesting that TBK-1 is involved in STAT3 Ser727 phosphorylation. Previous studies have shown that TBK-1 and its closely related kinase IKKε are involved in TLR-induced glycolytic reprogramming[6] and that deletion of TBK-1 in IKKε$^{-/-}$ macrophages reduced LPS-induced glycolysis that was dependent upon TRAF6 interaction suggesting that IKKε may play a role also in STAT3 phosphorylation.

We therefore pre-treated macrophages with the inhibitor BX-795, at 1 μM, at which concentration it is known to target both TBK-1 and IKKε[24,25], for 30 min prior to LPS stimulation. This pre-treatment reduced STAT3 Ser727 phosphorylation to background levels (Fig. 2e). These data suggest that the closely related kinases TBK-1 and IKKε are involved in STAT3 Ser727 phosphorylation. These findings are consistent with previous studies identifying interaction between overexpressed TRAF6 and TBK-1, and that these kinases may also play a role in downstream

**Fig. 2 TLR stimulation induces STAT3 pSer and mitochondrial localisation. a** Immortalised BMDMs (iBMDMs) ($1 \times 10^6$) were treated with IFNα (1000 U) for indicated times, **b** BMDMs ($1 \times 10^6$ cells/well) were treated with LPS for indicated times, and **c** BMDMs were also treated with TLR agonists, Pam$_3$Cys (TLR2, 100 ng/ml), Poly I:C (TLR3, 500 ng/ml), LPS (TLR4, 100 ng/ml), Loxoribine (TLR7, 10 mM), CpG-DNA (TLR9, 2.5 μM) and IFNα (IFNAR1, 1000 U) for 60 min. STAT3 phosphorylation status was detected by immunoblot of cellular lysates with anti-STAT3, anti-STAT3 pS727 and anti-STAT3 pY705 antibodies as indicated ($n = 3$). **d** MyD88$^{-/-}$/TRIF$^{-/-}$ iBMDM or **e** IFNAR1$^{-/-}$ BMDMs were treated with TLR ligands ($n = 2$) as described and immunoblot analysis with anti-STAT3 pS727 and anti-STAT3 antibodies of cellular lysates show ablated TLR-induced STAT3 Ser727 phosphorylation compared with IFNAR1-deficient BMDMs (Fig. 3e), but not IFNα ($n = 2$). (F) iBMDMs were treated with PBS or LPS (100 ng/mL) for 1 h and whole cell lysate, S100 or clean mitochondrial fractions isolated and separated by SDS-PAGE. Abundance of STAT3 and pS727 STAT3 in each fraction was observed as was fraction purity by probing with antibodies against Erk2 (S100), VDAC (mitochondria) or Lamin A/C (nucleus). Data shown are the mean ± S.E.M of **a** $n = 1$, **b** $n = 5$, **c, e** $n = 3$, **d, f** $n = 2$ independent experiments.

TLR signalling independent of their established role in mediating TLR-induced IRF3 phosphorylation and induction of IFNβ[26]. While PI3 kinase and Akt have previously been implicated in TLR-induced glycolytic reprogramming via TBK-1/IKKε[6], none of the seven Akt inhibitors screened in our study reduced LPS-induced mtROS (Fig. 3a). We therefore examined whether the PI3 kinase inhibitor TG100713 could also inhibit LPS-induced STAT3 Ser727 phosphorylation and noted that this inhibitor had no effect upon LPS-induced phosphorylation (Supplementary Fig. S2). Therefore, while Akt has previously been identified as a downstream target of TBK-1/IKKε in TLR-induced glycolysis in dendritic cells[6], neither Akt nor PI3 kinase were implicated in TBK-1-mediated glycolysis in macrophages[7] and do not appear to play a role in STAT3 S727 phosphorylation in macrophages. Together however, these studies identify a TRAF6, TBK-1/STAT3 signalling nexus leading to STAT3 S727 phosphorylation and mitochondrial localisation, providing a potential mechanism for the previously described role of TBK-1/IKKε signalling in TLR-induced glycolysis[6,7].

Indeed, while non-canonical TBK-1 recruitment and phosphorylation was identified as critical in macrophage LPS-induced glycolysis[7], the role of Akt in macrophage signalling was not established akin to dendritic cells[6]. Moreover, the role of these kinases may reflect cell specific signalling differences between macrophages and dendritic cells or reflect the kinetics of early and late glycolytic metabolic reprogramming pathways.

While the induction of glycolysis and metabolic reprogramming is increasingly recognised for its importance in the inflammatory state in macrophages, the mechanism of how TLRs promote this response is unknown. Our discovery identifies a molecular function for non-canonical kinase activity in TLR signalling via TBK-1 phosphorylation of STAT3, which subsequently translocates to the mitochondria.

**STAT3 pSer727 required for TLR-induced metabolic reprogramming.** Given that TBK-1 has been previously demonstrated to promote TLR-dependent glycolysis and that we have reported STAT3 Ser727-dependent metabolic changes, we hypothesised that the loss of STAT3 Ser727 phosphorylation would impede metabolic reprogramming. To address this, we performed a panel of metabolic analysis on primary peritoneal macrophages obtained from mice in which a serine to alanine (S727A) mutation was knocked into the endogenous STAT3 locus[27] abolishing Ser727 phosphorylation (STAT3 S727A; herein termed STAT3 SA). We used peritoneal exudate cells (PECs) for these studies because of the known requirement for STAT3 in growth factor signalling and differentiation (e.g. CSF-1, GM-CSF pathways)[28], moreover, PECs represent the local immunological environment, consisting predominately of myeloid cells[29]. We found that STAT3 SA PECs have a significant defect in both resting glycolysis and the early glycolytic burst observed in wild-type macrophages following LPS-challenge by assaying the ECAR (Fig. 4a). Together with the previous observations in TBK-1/IKKε-deficient

cells[7], these data support the concept that phosphorylation of STAT3 induces metabolic reprogramming.

To establish whether STAT3 Ser727 phosphorylation is necessary for mitochondrial reprogramming, we exposed WT and STAT3 SA peritoneal macrophages to LPS for 24 h and observed ablation of the ECAR and mitochondrial oxygen consumption rate (OCR) in STAT3 SA compared with WT cells (Fig. 4b, c). We next examined the functional capacity of the electron transport chain by determining the changes in OCR following sequential treatment of cells with mitochondrial ETC inhibitors. These data suggest that S727 phosphorylation of STAT3 is critical to elicit LPS-induced mitochondrial reprogramming and suggests STAT3 has parallel roles in LPS-induced metabolic reprogramming to that observed in tumour cells.

Consistent with what we have previously observed in cancer cells[11], the loss of STAT3 S727 phosphorylation leads to a diminished basal respiration rate. Moreover, STAT3 SA macrophages had a significant reduction in their maximal respiratory capacity compared with WT cells (Fig. 4d, e). These data mean whilst STAT3 SA PECs are viable and actively respiring, they are operating at close to their maximal capacity even in the absence of LPS stimulation. It should also be noted that the increase in basal respiration following a 24 h LPS-treatment of peritoneal macrophages we observe is the opposite of the LPS-mediated suppression of OCR observed in BMDMs, but is consistent with other studies on PECs[30,31]. This potentially reflects the different polarisation of these macrophage populations, where PECs display higher expression of M1 markers when compared with BMDMs[32]. Indeed, in addition to the intrinsic metabolic differences in macrophages from diverse microenvironments, it has been suggested that the process of culturing BMDMs for 7 days in M-CSF culturing may bias them towards M2 differentiation as compared with peritoneal macrophages[33]. Therefore, while both cell types are display an intrinsic capacity to induce a potent inflammatory response to challenge, their metabolic responses differ.

Macrophage activation by LPS is accompanied by a remodelling of the TCA cycle resulting in accumulating concentrations of TCA metabolites including succinate, which plays a critical role in mitochondrial reprogramming and inflammatory cytokine production. We found that WT peritoneal macrophages treated with LPS for 24 h displayed significantly increased succinate as expected. However, STAT3 SA macrophages failed to increase succinate concentrations as observed with wild type (WT) macrophages (Fig. 4f). Our data also show that STAT3 SA macrophages have lower basal ECAR, which is increased in response to LPS but not to the magnitude observed in WT macrophages. This is in line with previous observations that mitochondria from STAT3 SA cells are defective in ETC activity[10,11]. Thus, STAT3 SA mitochondria may be more reliant on aerobic glycolysis that is consistent with the significant increase in the lactate concentration in STAT3 SA macrophages in response to LPS stimulation (Fig. 4g). However, the lactate

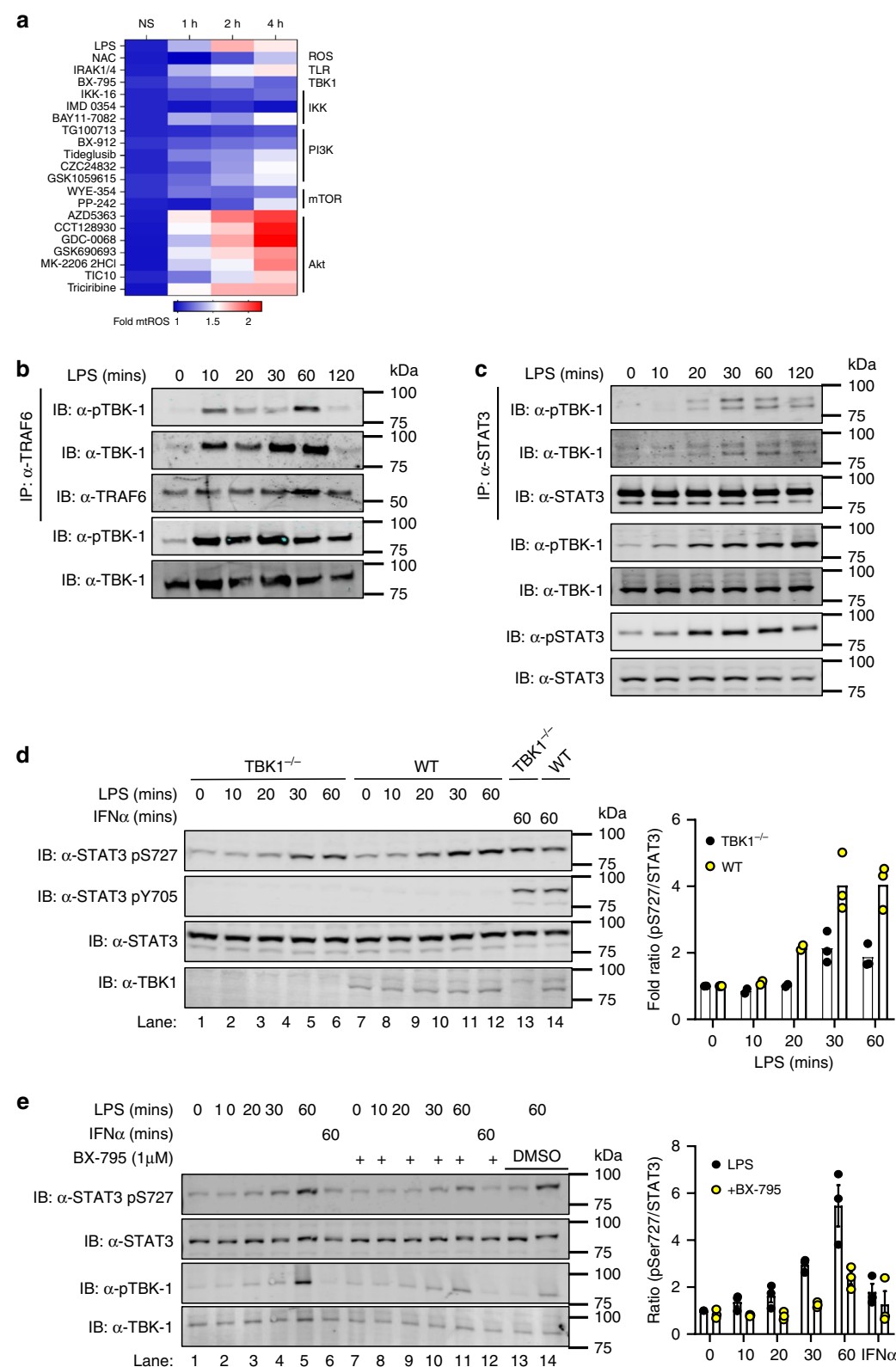

concentration in LPS-treated STAT3 SA macrophages only ever approaches the lactate concentration observed in unchallenged WT macrophages. In addition, we do not observe any increase in the lactate concentration in WT macrophages in response to LPS stimulation. Given that ECAR is typically associated with lactate production, this result appears somewhat counter-intuitive.

However, it is important to note that ECAR also measures the export of $CO_2$, hydration of $H_2CO_3$ and the dissociation to $HCO_3$ and $H^+$ from the respiratory chain which also contributes to the ECAR reading[34]. Together, these data show that LPS induced STAT3 Ser727 phosphorylation is required for TCA cycle (succinate concentration) and OXPHOS (OCR) augmentation.

**Fig. 3 TBK-1 interacts with and functions in STAT3 Ser727 phosphorylation. a** iBMDMs were seeded at $4 \times 10^4$ cells per well in triplicate 24 h prior to pretreatment with kinase inhibitors (Y-axis; 500 nM) for 30 min. MitoSOX was added for 10 min prior to challenge with LPS (100 ng/ml; 0, 1, 2 and 4 h). LPS-induced production of superoxide by mitochondria was analysed by measuring oxidised MitoSOX fluorescence at 580 nm. See also Supplementary Fig. 1. Cell lysates from LPS stimulated BMDMs (0–120 min) were immunoprecipitated with **b** anti-TRAF6 and **c** anti-STAT3 antibodies. TBK-1 interaction with TRAF6 and STAT3 was identified by immunoblot with anti-TBK-1 and anti-TBK-1 pS172 antibodies respectively. **b, c** Data represents three independent experiments with similar results. **d** TBK-1-deficient and WT macrophages were treated with LPS or IFNα for indicated times and STAT3 phosphorylation assayed in cellular lysates by immunoblot with indicated antibodies. TBK-1 expression was ascertained by immunoblot with anti-TBK-1 antibody. **e** BMDMs were pretreated with the TBK-1/IKKε (1 μM) inhibitor BX-795 for 60 min prior to LPS stimulation for indicated times. STAT3 and TBK-1 phosphorylation was observed by immunoblot with indicated antibodies. DMSO was added to lanes 13–14 as vehicle control. Immunoblot results are representative of three independent experiments and data presented (**d, e**) as mean ± S.E.M.

**IL-1β production after TLR4 challenge is STAT3 dependent.** Previous studies have emphasised that enhanced succinate production is a critical regulator of the pro-inflammatory response via ETC-mediated mtROS production, the expression of IL-1β[2,8,35]. Consistent with these observations we show that STAT3 SA macrophages generate significantly less IL-1β mRNA expression (Fig. 5a). We therefore, examined the kinetics and expression of cytokines in STAT3 SA PECs following LPS stimulation. Consistent with our mRNA data, while IL-1β expression increased steadily between 4 and 24 h of LPS challenge, IL-1β protein expression was significantly suppressed in STAT3 SA cell lysates compared with WT cells (Fig. 5b). Furthermore, TNF expression was reduced in STAT3 SA compared with WT cells (Fig. 5c), but did continue to increase parallel to WT expression, while IL-6 concentrations were only significantly different at 24 h post-LPS (Fig. 5d). Interestingly, whilst we observed increased IL-10 expression in unstimulated STAT3 SA macrophages, they did not respond to LPS with the increase in IL-10 production observed in WT cells (Fig. 5e). These results demonstrate that STAT3 Ser727 plays a crucial role in inflammatory cytokine expression following LPS challenge. Importantly, these findings are consistent with previous studies characterising metabolic reprogramming as integral to IL-1β production[2,8,35], while the temporal increase in cytokines such as IL-6 may be due to reduced autocrine induction due to reduced IL-1β or TNF production. Interestingly it also suggests that STAT3 Ser727 phosphorylation may play a role in steady state IL-10 expression.

Increased aerobic glycolysis in macrophages plays a critical role in disease pathogenesis during endotoxemia[8]. To investigate the role of STAT3 Ser727 phosphorylation in TLR-induced inflammation in vivo, we examined STAT3 SA mice in a model of LPS-induced sepsis. As can be seen in Fig. 5f-i, STAT3 SA mice demonstrate significantly reduced IL-1β, IL-6 and Ccl2, but not TNF, production in serum following acute LPS challenge, compared with WT mice. Consistent with our in vitro studies we observed no difference in IL-10 expression (Fig. 5j) which reflect the early response to LPS stimulation, whereas IL-10 expression may be delayed. Taken together, these results highlight the specificity of Ser727 phosphorylated STAT3 in mediating inflammatory gene induction.

Increasing evidence positions metabolic reprogramming as a key event in the inflammatory response following TLR activation. Importantly, this study identifies, for the first time, the molecular mechanism by which TLR signalling communicates with the mitochondria. We have demonstrated that TBK-1 contains a putative TRAF6 binding site allowing its direct recruitment to the TLR signalling pathway, facilitating TBK-1-mediated STAT3 Ser727 phosphorylation and translocation to the mitochondria. Macrophages unable to undergo STAT3 Ser727 phosphorylation display impaired TLR-induced glycolytic reprogramming, reduced pro-inflammatory metabolite production and diminished inflammation. This discovery parallels observations made in cancer cells, which further enhances the concept that inflammatory macrophages induce the Warburg effect to engender a pro-inflammatory phenotype. As such, STAT3 is not only a central immune regulatory transcription factor, but additionally rapidly orchestrates innate immune mitochondrial reprogramming and inflammatory cell metabolism via non-canonical signalling.

## Methods

**BMDM cell culture.** WT, IFNAR1$^{-/-}$ (kind gift from Prof Paul Hertzog, Hudson Institute of Medical Research), TBK-1 $^{F/F}$, TBK-1 $^{F/F}$/Vav-iCre BMDMs were differentiated in DMEM containing 30% M-CSF conditioned media obtained from supernatants of L929 fibroblasts, centrifuged to remove cell debris (5 min, 300 g) and filtered through a 0.22 μm filter. Leg bones of mice were surgically removed and cleaned bones were cut with scissors and flushed with sterile PBS via a syringe. Bone marrow suspension was passed through a 70 μm cell strainer to remove clumps and cells cultured in low-adherence 10 cm tissue culture plates in L929 supplemented 10% FCS, DMEM (Gibco) with added L-glutamine (Gibco) at 37 °C, 5% CO$_2$ for 7 days. Cells were supplemented with a further 5 ml of L929 conditioned FCS/DMEM on day 3. Cells were removed from tissue culture plates with gentle scrapping and seeded at desired densities in 1% FCS/DMEM, supplemented with L-glutamine 24 h prior to stimulation or treatment.

Immortalised WT and MyD88$^{-/-}$ TRIF$^{-/-}$ BMDMs (iBMDMs) were generated from indicated mice with J2 recombinant retrovirus carrying v-myc and v-raf oncogenes[36,37].

Immortalised WT BMDMs were a kind gift of Prof Douglas Golenbock (UMASS). HEK293T cells were obtained from ATCC were grown in 10% FCS in DMEM supplemented with L-glutamine and grown in humidified 5% CO$_2$ at 37 °C.

**Mice.** All experimental procedures were approved by the Monash Medical Centre Animal Ethics Committee. 6–12 weeks of age female and male mice were used. STAT3 S727A (STAT3 SA)[27], IFNAR1$^{-/-}$ and C57BL/6J WT were maintained at the Monash Medical Centre Animal Facility under specific pathogen-free conditions in accordance with Australian Government animal welfare regulations. TBK-1 $^{F/F}$ and TBK-1 $^{F/F}$/Vav-iCre were maintained at the UMass Medical School Animal Facility in compliance with the federal regulations set forth in the Animal Welfare Act, the recommendations in the Guide for the Care and Use of Laboratory Animals of the National Institutes of Health, and the guidelines of the UMass Medical School Institutional Animal Use and Care Committee.

**Peritoneal macrophage isolation.** Peritoneal cells were collected from WT and STAT3 S727A (STAT3 SA) mice via peritoneal lavage with 5 ml of cold sterile PBS supplemented with 5 mM EDTA. Cells were centrifuged (300 g, 5 min) and allowed to adhere to tissue culture plates for 1 h in 10% FCS, DMEM (Gibco) and 2 mM L-glutamine (Gibco). Cells were washed three times with sterile PBS after 180 min to remove non-adherent cells and grown overnight at 37 °C, 5% CO$_2$ in 1% FCS/DMEM supplemented with 2 mM L-glutamine prior to experimentation[29].

**Immunoprecipitation of TRAF6 and STAT3.** To identify the binding interface between STAT3 and TRAF6 we generated a panel of STAT3 mutants by site directed mutagenesis of E28, E100, E680 or E28 and E680 using the following primers E28A: F 5′- CAGTGACAGCTTCCCATGGCGCTGCGGCAGTTTC; R 5′- GAAACTGCCGCAGCGCCATGGGAAGCTGTCACTG, E100A: F 5′- CTTGA GAAGCCAATGGCGATTGCCCGGATTGTG; R 5′- CACAATCCGGGCAATCG CCATTGGCTTCTCAAG, E680A: F 5′-CACAATCCGGGCAATCGCCATTGGC TTCTCAAG, R 5′- CTTGAGAAGCCAATGGCGATTGCCCGGATTGTG. The resultant FLAG-tagged STAT3 constructs were transfected into 293T using lipofectamine 3000 (Thermo Fisher Scientific, Cat# L3000001). 48 h after transfection cells were lysed (50 mM Tris, pH 7.4, 1.0% Triton X-100, 150 mM NaCl, 1 mM EDTA, 2 mM Na$_3$VO$_4$, 10 mM NaF, 1 mM PMSF and protease cocktail inhibitor (Roche) and centrifuged at 18,000 g for 5 min (4 °C) to remove debris. GST tagged TRAF6 C-domain was cloned into pGEX-4T-3 and expressed in BL21(DE3)

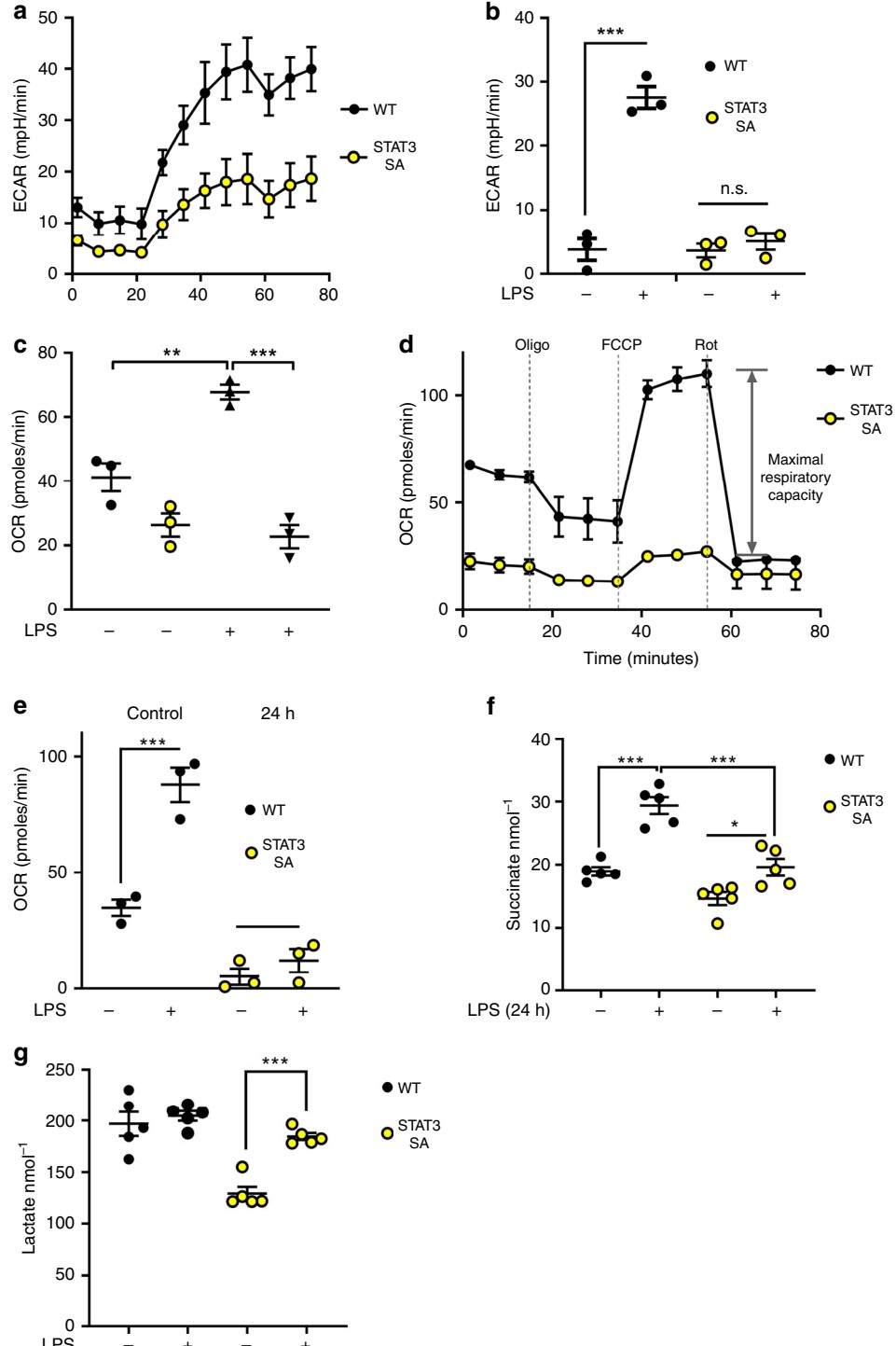

**Fig. 4 STAT3 pSer727 is required for LPS-induced metabolic reprogramming. a** Peritoneal macrophages (1 × 10⁵/well) obtained from WT vs STAT3 SA mice ($n = 3$ per genotype). Real-time changes in the extracellular acidification rate (ECAR) of WT and STAT3 SA macrophages treated with LPS (**b, c**) WT and STAT3 SA peritoneal macrophages were stimulated with LPS or vehicle as indicated for 24 h. **b** ***$p < 0.0001$, and **c** **$p = 0.0034$, ***$p < 0.0001$. **a, b** The ECAR and oxygen consumption rate (OCR) were analysed as indicators of oxidative phosphorylation and glycolysis, using a Seahorse XFp analyser. **d** Peritoneal macrophages were assayed for real-time changes in the OCR by sequential treatment with sequential treatment of cells with electron train chain inhibitors (oligomycin: adenosine triphosphate (ATP) synthase inhibitor; FCCP: H+ ionophore cyanide p-triflurmethooxyphenyl-hydrozone; and Complex I inhibitor rotenone). **e** Comparison of maximal respiratory capacity (MRC) between WT and STAT3 SA peritoneal macrophages stimulated with LPS or vehicle for 24 h. Results presented (**a–e**) as the mean ± S.E.M of peritoneal macrophages obtained from three individual mice per genotype, ***$p = 0.0004$, One-way ANOVA with Tukey's multiple comparisons test (**f–g**) Peritoneal macrophages obtained from individual mice were seeded at 1 × 10⁵ cells/well and stimulated with LPS (100 ng/ml) for 24 h prior to analysis of metabolites succinate and lactate produced in WT and STAT3 SA cultured supernatants. Data presented as mean values of duplicate technical replicates ± SEM of five individual mice per genotype, **f** *$p = 0.0292$, ***$p < 0.0001$ **g** ***$p = 0.0004$, One-way ANOVA with Tukey's multiple comparisons test.

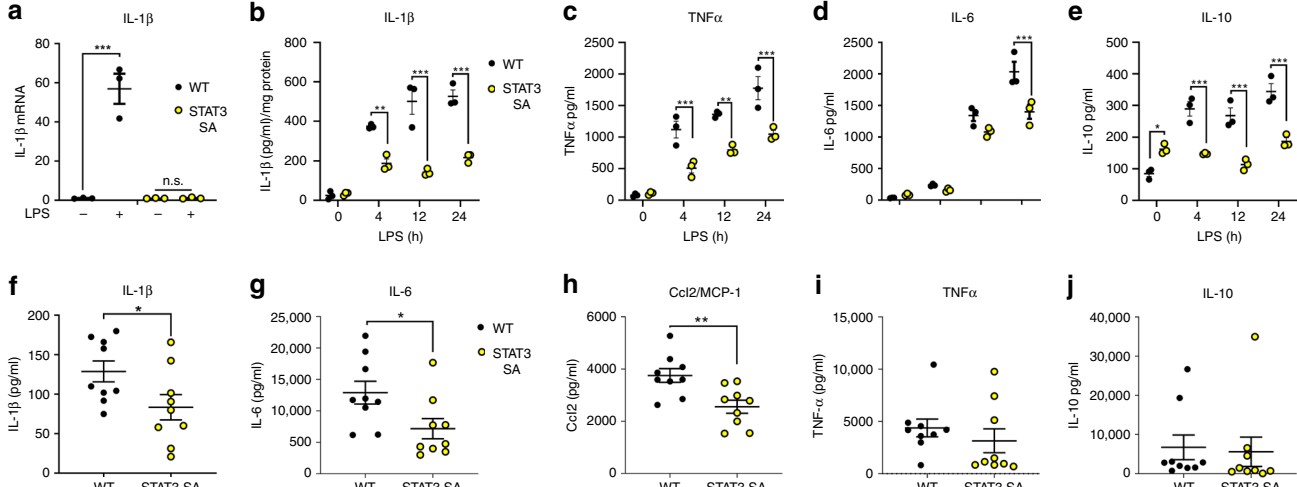

**Fig. 5 STAT3 pS727 is required for LPS-induced cytokine expression. a** Peritoneal macrophages obtained from three mice per genotype were seeded at $5 \times 10^5$ cells/well in and stimulated with LPS (100 ng/ml) for 24 h prior to analysis IL-1β mRNA expression. Data are presented as mean values ± SEM, ***$p < 0.0001$, n.s.—not significant, one-way ANOVA, Tukey's multiple comparisons test. Peritoneal macrophages generated from individual WT and STAT3 mice (3 mice per genotype) were seeded at $1.5 \times 10^5$ in triplicate. Macrophages were stimulated for 4, 12, and 24 h with 100 ng/ml of LPS and **b** cell lysates assayed for IL-1β and presented as IL-1β/mg of total protein **$p = 0.0011$, ***$p < 0.0001$, and cultured supernatants assayed for **c** TNF (4 h, ***$p = 0.0007$; 12 h **$p = 0.0015$; 24 h ***$p = 0.0001$), **d** IL-6 (***$p < 0.0001$), and **e** IL-10 (*$p = 0.0168$, ***$p < 0.0001$) by ELISA. Data presented are the mean of triplicate values from three individual mice ($n = 3$) per genotype per time point, Analyses 2 way ANOVA, Sidak's multiple comparison tests. WT and STAT3 SA mice ($n = 9$ mice per genotype) were intraperitoneally treated with 10 mg/kg of LPS for 90 min and serum analysed for **f** IL-1β (*$p = 0.0444$), **g** IL-6 (*$p = 0.0313$), **h** MCP-1 (**$p = 0.0046$), **i** TNF ($p = 0.3994$) and **j** IL-10 ($p = 0.8204$) protein expression by ELISA or cytometric bead array. Data are presented as mean values ± SEM, Students unpaired $t$ test, two-tailed.

bacteria (Thermo Fisher Scientific, Cat# EC0114) as described previously[20]. Briefly, TRAF6 transformed BL21 bacteria were cultured in Luria Broth supplemented with ampicillin (100 μg/ml) and when an OD600 of 0.6 reached TRAF6 expression was initiated by incubation with 500 μM Isopropyl β- d-1-thiogalactopyranoside for 16 h at 16 °C. Bacteria was pelleted by centrifugation and lysed by incubation in 50 mM Tris, 150 mM NaCl at pH 7.6 with complete protease inhibitors (Roche) (10 ml per g of bacteria) and sonicated at 4 °C for 30 s bursts with 30 s between for a total of 10 cycles. Bacterial lysate was clarified by centrifugation at $40,000 \times g$ for 45 min at 4 °C and the clarified protein bound to glutathione agarose beads (Thermo Fisher Scientific, Cat# 25236) at 4 °C for 4 h with rotation. Beads were washed in 10x bead volume of lysis buffer three times. Protein induction and TRAF6-bound beads were assessed by SDS-PAGE and Coomassie blue staining of a protein of ~63 kDa. To observe STAT3 binding to recombinant TRAF6, 20 μl of GST-TRAF6 beads were added to equivalent, clarified lysate from transfected 293T cells described above. Binding was performed at 4 °C for 4 h with rotation. Beads were washed in 10x bead volume of buffer 50 mM Tris, pH 7.4, 1.0% Triton X-100, 150 mM NaCl, 1 mM EDTA, 2 mM $Na_3VO_4$, 10 mM NaF, 1 mM PMSF (all Sigma Aldrich) and protease cocktail inhibitor protease (Roche, Cat# 05892970001)) three times for 5 min each with rotation. After the final wash supernatant was removed and beads resuspended in Laemmli buffer, boiled for 5 min and, together with input protein sample separated by SDS-PAGE. Gels were transferred to nitrocellulose, blocked in odyssey blocking buffer (Li-cor, Cat#927–70001) and incubated with indicated primary antibodies. Equivalent GST-TRAF6 loading was detected by Coomassie blue staining of SDS-PAGE gels following transfer.

**Immunoblot**. To detect the phosphorylation status of STAT3, BMDMs ($1 \times 10^6$) cells were seeded in 6-well plates 24 h prior to stimulation with agonists (IFNα, 1000U; kind gift Prof Paul Hertzog, Hudson Institute of Medical Research, Pam₃Cys, 100 ng/ml; EMC microcollections, Cat#L2000S), poly I:C, 10 μg/ml; Invivogen, Cat# tlt-picw, LPS, 100 ng/ml; Invivogen Cat# tlrl-b5lps, Loxoribine, 500 μM, Invivogen, Cat# TLRl-lox and CpG-DNA 1668S, 500 nM, custom synthesis Geneworks) for indicated times. pBMDMs were lysed in a modified RIPA lysis buffer [50 mM Tris (pH 7.4), 150 mM NaCl, 1 mM EDTA, 1% (v/v) Triton X-100, 0.3% (w/v) sodium deoxycholate, 0.3% (w/v) SDS (all Sigma Aldrich) supplemented with protease (Roche, Cat# 05892970001) and phosphatase (Roche, Cat# 04906845001) inhibitors. Protein concentration was assayed using the *DC* Protein Assay (Bio-Rad, Cat# 5000111). Samples were reduced and boiled in Laemmli buffer containing 10% (v/v) beta-mercaptoethanol, resolved by SDS-PAGE, transferred onto PVDF (EMD Millipore, Cat# IPFL00010), blocked in Odyssey Blocking Buffer (Li-cor, Cat#927-70001)and incubated with the following primary antibodies overnight: pY705-STAT3 (1:1000, Cat# 9131), pS727-STAT3 (1:1000, Cat#

9134), STAT3 (124H6; 1:1000, Cat# 9139), α-TBK-1 (1:1000, Cat# 3013), α-TBK-1 pSer172 (D52C2; 1:1000, Cat# 5483) were sourced from Cell Signaling Technology. TRAF6 antibody was from Santa Cruz Biotechnology (d-10; 1:500, Cat# SC-8409) Membranes were then probed with the appropriate IRDye conjugated secondary antibodies: anti-rabbit IgG (1:5000, Thermo Fisher Scientific, Cat#A21076), anti-mouse IgG (1:5000, Rockland Immunochemicals, Cat#610-445-002), anti-rat IgG (1:5000, Rockland Immunochemicals, Cat# 612-145-002). Membranes were scanned using an Odyssey® Infra-red Imaging System. To assess the role of kinases in STAT3 phosphorylation, BMDMs were pretreated with either 1 μM BX-795 (Caymen Chemicals, Cat# 14932) or 1 μM TG10073 (Selleck Chemicals, Cat#S2870) for 60 min prior to LPS challenge and assayed as described above. Full figure immunoblots for all figures presented are provided in the Supplementary Information.

**Mitochondria isolation**. Pure mitochondria were isolated from iBMDM[11]. $5 \times 10^7$ – $1 \times 10^8$ iBMDMs were collected by scraping with a rubber policeman, washed twice and resuspended in 5× the pellet volume of buffer A (20 mM HEPES pH 7.6, 220 mM mannitol, 70 mM sucrose, 1 mM EDTA, 0.5 mM PMSF and 2 mg/ml bovine serum albumin]. Cells were incubated on ice for 15 min to facilitate cell swelling before being subjected to nitrogen cavitation under 200 PSI of pressure for 5 min (PARR instruments, Cat# 4639). Cell homogenate were centrifuged at $800 g$ for 10 min at 4 °C and the mitochondria containing supernatant retained and centrifuged at $10,000 g$ for 20 min at 4 °C. The supernatant representing the crude mitochondrial fraction was resuspended in 1 ml Solution B (20 mM HEPES pH 7.6, 220 mM mannitol, 70 mM sucrose, 1 mM EDTA, 0.5 mM PMSF) and loaded on top of a stepwise Percoll (GE Healthcare, Cat# 17-0891-01) gradient comprised of 1 mL 80% Percoll/balance solution A, 4.5 ml 56% Percoll/balance solution A and 4.5 ml 23% Percoll/balance solution A. Gradients were centrifuged at $65,000 g$ for 45 min and mitochondria isolated from the junction of the 56% and 23% layers. Mitochondria were washed twice in solution B and mitochondrial protein content detected by adding 1 μl of mitochondrial suspension to 600 μl of 50 mM Tris pH 7.4, 0.1% (w/v) SDS and measuring the absorbance at 280 and 310 nm. Mitochondrial protein concentration in mg/ml is given by $(A_{280nm} - A_{310nm})/1.05 \times 600$.

S100 fractions were prepared by collecting the supernatant after the initial $10,000 g$ crude mitochondrial isolation centrifugation which was centrifuged at $100,000 g$ in a Beckman benchtop ultracentrifuge. Protein concentration was characterised by Bradford assay (Thermo Fisher, Cat# 23236). Equivalent protein was resolved through SDS-PAGE and protein content and fractionation purity related by western blot with antibodies against the indicated proteins (α-VDAC D73D12, 1:1000, Cat# 4661; α-Lamin A/C, 1:1000, Cat#2032; Cell Signaling Technologies; α-ERK1/2 K-23, 1:1000, Santa Cruz Biotechnology, Cat# SC-94).

**Gene expression and cytokine analysis**. Total RNA from WT and STAT3 SA peritoneal macrophages was isolated using the RNAeasy Isolation Kit (Qiagen, Cat# 74104) and reverse transcribed with random hexamers (Thermo Fisher Scientific, Cat# N8080127) using Moloney murine leukaemia virus reverse transcriptase (Promega, Cat# 1701) according to manufacturers' instructions. mRNA was quantified with SYBR reagents (Thermo Fisher Scientific, Cat# A46012) using primer pairs targeting il1b (Forward 5′- CAACCAACAAGTGAT ATTCTCCATG- 3, Reverse 5′- GATCCACACTCTCCAGCTGCA- 3′), and 18S (Forward 5′- GTAACCCGTTGAACCCATT- 3′, Reverse 5′- CGAATCGAATCG GTAGTAGCG- 3′). Relative mRNA expression was analysed using the comparative CT method, normalising genes of interest to the 18S housekeeping gene and fold gene induction calculated relative to expression in control samples.

Quantification of cytokines secreted from peritoneal macrophages and serum, and peritoneal macrophages cellular lysates were conducted using ELISA kits from R&D Systems (IL-1β: Cat# DY401, TNF: Cat# DY410, IL-6: Cat# DY406 and IL-10: Cat# DY417) or Cytokine Bead Array Mouse Inflammation kit (BD Biosciences, Cat# 552364) according to manufacturers' instructions.

**Monitoring mitochondrial ROS production**. iBMDMs ($4 \times 10^4$/ml) were seeded in triplicate in black microtest Optilux 96-well plates (BD Falcon) in 1% FCS/ DMEM (Gibco) and L-glutamine (Gibco) media for 4 h and then replaced with phenol-red free 1% FCS/DMEM (Gibco) and L-glutamine (Gibco) 20 h prior to 30 min pretreatment or not with kinase inhibitors (500 nM). Macrophages were incubated with a final concentration of 1 μM MitoSox (Thermo Fischer Scientific, Cat# M7512) for 10 min before treatment with LPS (100 ng/ml, Invivogen). Plates were analysed for fluorescence emission as a marker of mtROS production (excitation/emission 510/580 nm) every 60 min for 4 h (ClarioStar Plate Reader, BMG) and expressed as the fold induction compared with nontreated control macrophages.

**Seahorse assay and metabolite analysis**. The ECAR and OCR of WT and STAT3 SA peritoneal macrophages were measured with a Seahorse XF Analyser. To measure ECAR and OCR in real-time, cells were isolated from mice, seeded at $1 \times 10^5$ cells/ well in eight-well miniplate format and allowed to adhere overnight. Cells were washed twice in media and treated with LPS (100 ng/ml) for 24 h. One hour prior to reading cells were washed twice with, and then cultured in Seahorse XF base medium (Agilent Technologies) supplemented with 1 mM pyruvate, 2 mM L-glutamine and 10 mM glucose in an incubator without $CO_2$. ECAR and OCR were measured under basal conditions prior to sequential treatment of cells with electron train chain inhibitors 1 μM oligomycin, 1.5 μM FCCP-cyanide p-tribluromethoxyphenyl-hydrazone, and 1 μM antimycin A and rotenone (Seahorse XF Cell Mito Stress Test kit, Agilent, Cat# 103015–100). In experiments examining real-time induction of ECAR by LPS, cells were isolated and seeded as described above, however following basal measurement of ECAR, cells were challenged with LPS (100 ng/ml) and ECAR measured for 180 min. Data represent mean ± SEM of triplicate wells from at least three independent mice.

Cultured supernatants from LPS-treated WT and STAT3 SA peritoneal macrophages were also assayed by Succinic Acid Colormetric Assay Kit (Biovision, Cat# K649) and Lactate-Glo Assay (Promega, Cat# J5021) for succinate and lactate metabolite concentrations respectively according to manufacturers' instructions.

**LPS-induced model of sepsis**. Sepsis was induced in male and female WT and STAT3 SA mice (aged 6–14 weeks) following i.p. injection with 10 mg/kg in a total volume of 100 μl of LPS (E. coli 055:B5 ultrapure; Invivogen). Mice were culled after 90 min and serum collected via cardiac puncture for measurement of serum cytokines.

**Quantification and statistical analysis**. Statistical analyses were conducted using specific statistical tests as indicated in the figure legends using GraphPad 8.01 software for each experiment. Data are represented as the mean ± standard error of the mean as indicated in the figure legends which includes the biological and experimental replicates. Significance is depicted with asterisks on graph as follows: $*p < 0.05$, $**p < 0.01$ and $***p < 0.001$.

**Reporting summary**. Further information on research design is available in the Nature Research Reporting Summary linked to this article.

## Data availability

The authors declare that the data supporting the findings in this study are available within the manuscript and Supplementary Information or from the corresponding author upon request. Source data are provided with this paper.

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

## Acknowledgements

This work was supported by the Victorian State Government Operational Infrastructure Scheme. D.J.G. is supported by a mid-career fellowship from the Victorian Cancer Agency (MCRF19033) and grants from the United States Department of Defence (CA150132) and the Cancer Council Victoria (GNT1145028). H.A. is supported by a scholarship from the College of Applied Medical Sciences, Shaqra University. The authors also wish to thank Dr Rebecca Smith with editorial assistance in preparing this manuscript.

## Author contributions

A.M., D.J.G. and E.L. conceived and designed the concept and experiments. J.B., H.A., K.L., F.J.K., W.S.N.J., F.M., D.J.G., N.B. and D.J.G. designed, conducted and interpreted experiments. D.D.N., C.L., K.A.F., F.H. and E.L. contributed reagents and the manuscript was written and edited by A.M. and D.J.G.

## Competing interests

The authors declare no competing interests.
