## [Peer Review File · Nature Communications]

Reviewers' comments:

Reviewer #1 (Remarks to the Author):

In the manuscript "STAT3 Serine phosphorylation by TBK-1 is required for TLR4 metabolic reprogramming" by Balic et al, the authors make the very interesting finding that STAT3 is a regulator of LPS-induced metabolism in macrophages. This area of biology is attracting an increasing amount of attention and the mechanisms proposed are important for the field. After performing the suggested experiments, I would be supportive of publication.

1. There are some textual adjustments that should be made. The first sentence of the results references myddosome assembly upon TLR activation. The references used refer to studies where only death domains were examined in cell free assays. The authors are encouraged to add primary references for studies that monitored myddosome assembly in macrophages. The authors should also discuss and cite the studies of PI3Kinase and AKT in TLR mediated metabolic programming. Otherwise, the reader may get the impression that STAT3 is the only regulator of metabolism. How all of these regulators interact functionally should be a point of discussion.

2. The conclusions from Figure 1D are important for this study, as they indicate a direct link between the myddosome and STAT3. However, confidence in this conclusion is lessened by the poor quality of these blots. The blots in Figure 1D need to be improved.

3. Figure 1E needs to be complemented with microscopy for other organelles in the cell. For example, lysosomes or endosomes could be examined for specificity of STAT3 localization to mitochondria. An additional concern is the quality of the imaging presented in Sup Figure 1G. I am not convinced that these images shown TRAF6 on mitochondria. I am also concerned about the biochemical fractionation data used to complement the microscopy. The authors show that TRAF6 is present on mitochondria at all times, yet West and Ghosh (referenced in this manuscript) showed LPS-inducible recruitment of TRAF6 to mitochondria. One possible explanation for these disparate findings is that the mitochondrial preparations used here might be contaminated with other organelles. The authors are encouraged to perform more control experiments and perhaps complementary fractionations procedures to illustrate this point. With all these statements made, the authors should not be disheartened, as the study of inducible protein transport in the innate immune system is very difficult. In fact, I consider these experiments to be unnecessary for the study and would be happy if all of this cell biological analysis was removed. The data following the microscopic and fractionation analyses is more compelling and leads to a cleaner narrative.

4. The TRAF6-TBK1 interaction described in Figure 2 bolsters data recently published by Tan and Kagan, where they showed that TRAF6 KO cells are defective for TBK1 recruitment and activation. The authors should discuss this work.

5. While the authors should be commended for the quality of the metabolic data presented in Figure 3, which nicely illustrates the importance of STAT3 in LPS induced metabolism, there is one point of confusion. The authors demonstrate that STAT3 SA cells are defective for ECAR increases (Figure 3A and Supp Figure 3). Yet they show that STAT3 SA cells have increased lactate production (Figure 3F). How is this possible, since lactate is what causes ECAR changes? Perhaps the answer relates to the fact that the increases in lactate production observed in STAT3 SA cells only approach the amount of lactate in the media of unstimulated Wild Type cells. In other words, the delta is not as important as the amount of lactate. Some discussion on this point would be helpful.

6. The data on IL-1 expression and protein abundance in macrophages (Figure 3) should be complemented by kinetic analyses and westerns. Why is IL-10 not an inducible protein in Figure 3J?

Reviewer #2 (Remarks to the Author):

Balic and colleagues have investigated the role of STAT3 in TLR-mediated metabolic reprogramming by macrophages. Their data point towards a key role for Ser727 phosphorylation of STAT3 in glycolytic and mitochondrial rewiring downstream of TBK1. The finding that STAT3 may form a link between TLR4 signaling and metabolic reprogramming is novel, but incremental and the current study is still pretty premature and requires a substantial amount of additional (cross)validation to be able to fully support the conclusions that are currently drawn by the authors.

My main points of critique are the following:

- 1) Based on the presence of binding motifs and IP assays, the authors conclude that TBK1 and STAT3 directly interact with TRAF6. This evidence is rather circumstantial. Additional experiments should be performed in which those binding motifs are mutated to see if the 'interaction' is lost.
- 2) The data regarding a direct role for TBK1 in phosphorylating STAT3 are rather preliminary as they are based on inhibiting TBK1 using BX795, an inhibitor known to have some off-target effects. The inhibitor data should be validated in TBK1 silenced macrophages and in which TBK1 has a mutated TRAF6-binding motif.
- 3) In general, WB signals should be quantified (phospho signal over total protein signal) as in several blots, the differences in signal are quite subtle.
- 4) In figure 1f, VDAC signal should be shown in the cytosolic fraction and tubulin in the mitochondrial fraction. Now it is unclear whether the fractions are pure or not.
- 5) TBK1 has been shown to promote early glycolytic programming through Akt-driven HKII activation in dendritic cells (everts et al, 2014). Is this a route parallel to STAT3 through which TBK1 regulates metabolic reprogramming? What is the relative contribution? This is currently unclear. This should be tested, for instance are there additional metabolic effects TBK1 is inhibited/silenced in STAT3 SA cells?
- 6) The authors state the 'data clearly show that S727 phosphorylation of STAT3 is critical to LPS-induced metabolic programming'. However, baseline mitochondrial metabolism is already significantly impaired in STAT3 SA cells. This makes it impossible to conclude that STAT3 is required for LPS-driven mitochondrial reprogramming, as these STAT3 SA cells seem to already have 'crippled' mitochondria.
- 7) In Fig S3 the authors find that STAT3 SA cells have impaired baseline and LPS-driven ECAR. It remains unclear how STAT3 would regulate ECAR (glycolysis). Or is in this case the difference in ECAR a reflection of reduced mitochondrial activity (hence CO₂ driven acidification)?
- 8) The authors use HIF1a mRNA as a readout for inflammatory responses down stream of mitoROS. The papers they refer to show that ROS promotes HIF1a protein stabilization, not mRNA increase. The authors should assess HIF1a protein.
- 9) Can the authors speculate how STAT3 is recruited to TRAF6? It already seems to be present in the traf6 complex (Fig 1a) before it is phosphorylated (Fig1b).
- 10) The colorcoding of Fig 2a is not clearly indicated
- 11) In Figure 3 the authors show that LPS enhances OCR. Yet, there are numerous studies showing that LPS leads to a reduction in mitochondrial respiration in myeloid cells over time. How can this be explained?

Reviewer #1:

1. *There are some textual adjustments that should be made. The first sentence of the results references myddosome assembly upon TLR activation. The references used refer to studies where only death domains were examined in cell free assays. The authors are encouraged to add primary references for studies that monitored myddosome assembly in macrophages.*

We have added primary references to the section regarding Myddosome formation in macrophages quoting primary research articles: Zaroni et al, Cell, 2011; Latty et al, eLife, 2018; Tan and Kagan, Cell, 2019; Rosadini et al, Cell Host Microbe, 2015; De Nardo et al, JBC, 2018.

The authors should also discuss and cite the studies of PI3Kinase and AKT in TLR mediated metabolic programming. Otherwise, the reader may get the impression that STAT3 is the only regulator of metabolism. How all of these regulators interact functionally should be a point of discussion.

We have added introductory and discussion text regarding the role of other kinases such as PI3 kinase and Akt and how these kinases may interact functionally and their impact upon TLR-induced glycolytic reprogramming in different cell types.

We also included the data related to the PI3 kinase inhibitor TG100713 into the main text and that Akt inhibitors were not identified in our initial kinase screen (Figure 3).

Page 9: . While PI3 kinase and Akt have previously been implicated in TLR-induced glycolytic reprogramming via TBK-1/IKK ϵ (Everts et al, 2014), none of the seven Akt inhibitors screened in our study reduced LPS-induced mtROS (Fig 3a). We therefore examined whether the PI3 kinase inhibitor TG100713 could also inhibit LPS-induced STAT3 Ser727 phosphorylation and noted that this inhibitor had no effect upon LPS-induced phosphorylation (Supplemental Data 1b). Therefore, while Akt has previously been identified as a downstream target of TBK-1/IKK ϵ in TLR-induced glycolysis in dendritic cells (Everts et al, 2014), neither Akt nor PI3 kinase were implicated in TBK-1-mediated glycolysis in macrophages (Tan and Kagan, 2019) and do not play a role in STAT3 S727 phosphorylation in macrophages.

and

Indeed, while non-canonical TBK-1 recruitment and phosphorylation was identified as critical in macrophage LPS-induced glycolysis (Tan and Kagan, 2019), the role of Akt in macrophage signalling was not established akin to dendritic cells (Everts et al, 2014). Moreover, the role of these kinases may reflect cell specific signalling differences between macrophages and dendritic cells or reflect the kinetics of early and late glycolytic metabolic reprogramming pathways.

We hope this expanded description of kinases such as Akt and PI3 kinase is acceptable to the reviewer.

2. *The conclusions from Figure 1D are important for this study, as they indicate a direct link between the myddosome and STAT3. However, confidence in this conclusion is lessened by the poor quality of these blots. The blots in Figure 1D need to be improved.*

We have replaced Figure 1d with two new immunoblots that no constitute Figures 2e and 2f (complemented with densitometry as suggested by Reviewer #2). We found that immortalized macrophages display constitutive STAT3 S727 phosphorylation, due we believe to increased sensitivity to growth factor STAT3 signaling as a requirement for immortalization. Unfortunately, we could not source MyD88/TRIF double-deficient mice and had to use our immortalised MyD88/TRIF^{-/-} macrophages, and we could not decrease the background further than is shown in Fig 2d.

3. *Figure 1E needs to be complemented with microscopy for other organelles in the cell. For example, lysosomes or endosomes could be examined for specificity of STAT3 localization to mitochondria. An*

additional concern is the quality of the imaging presented in Sup Figure 1G. I am not convinced that these images shown TRAF6 on mitochondria. I am also concerned about the biochemical fractionation data used to complement the microscopy. The authors show that TRAF6 is present on mitochondria at all times, yet West and Ghosh (referenced in this manuscript) showed LPS-inducible recruitment of TRAF6 to mitochondria. One possible explanation for these disparate findings is that the mitochondrial preparations used here might be contaminated with other organelles. The authors are encouraged to perform more control experiments and perhaps complementary fractionations procedures to illustrate this point. With all these statements made, the authors should not be disheartened, as the study of inducible protein transport in the innate immune system is very difficult. In fact, I consider these experiments unnecessary for the study and would be happy if all of this cell biological analysis was removed. The data following the microscopic and fractionation analyses is more compelling and leads to a cleaner narrative.

We agree with the reviewer that the microscopy data was insufficient to draw clear conclusions and appreciate the reviewer considering the data unnecessary due to the strength of our metabolic data. We have therefore removed the microscopy data from the manuscript. However, we did conduct new fractionation of LPS stimulated immortalised BMDMs (new Fig 2f) as we felt it important to demonstrate enrichment of mitochondrial STAT3. This method requires large numbers of macrophages (~ 2x10⁸/treatment), which for feasibility reasons meant that we chose to use iBMDMs. These cells have higher basal pS727 STAT3 than PECs as observed in lane 1 (untreated Lysate sample). Importantly however, both the western blot image and densitometry analysis demonstrates a clear enrichment of mitochondrial STAT3 following LPS challenge.

Page 6: We therefore, performed biochemical fractionation of LPS-treated macrophages (Fig. 1f) and demonstrate an enrichment of STAT3 pSer727 in mitochondrial fractions after 60 mins LPS stimulation.

4. The TRAF6-TBK1 interaction described in Figure 2 bolsters data recently published by Tan and Kagan, where they showed that TRAF6 KO cells are defective for TBK1 recruitment and activation. The authors should discuss this work.

As requested, we have included further discussion of our results within the context of the studies by Tan and Kagan with reference to the role of TRAF6 in both studies.

Page 7: This is consistent with the recent publication from Tan and Kagan (2019) who showed that TRAF6 depleted macrophages are defective for TBK-1 recruitment to the Myddosome and induction of TLR glycolysis.

Page 9: Therefore, while Akt has previously been identified as a downstream target of TBK-1/IKK ϵ in TLR-induced glycolysis in dendritic cells (Everts *et al*, *Nat Immunol*, 2014), neither Akt nor PI3 kinase were implicated in TBK-1-mediated glycolysis in macrophages (Tan and Kagan, *Cell*, 2019) and do not play a role in STAT3 S727 phosphorylation in macrophages. Together however, these studies identify a TRAF6, TBK-1/STAT3 signalling nexus leading to STAT3 S727 phosphorylation and mitochondrial localisation, providing a potential mechanism for the previously described role of TBK-1/IKK ϵ signalling in TLR-induced glycolysis (Everts *et al*, *Nat Immunol*, 2014; Tan and Kagan, *Cell*, 2014)

5. While the authors should be commended for the quality of the metabolic data presented in Figure 3, which nicely illustrates the importance of STAT3 in LPS induced metabolism, there is one point of confusion. The authors demonstrate that STAT3 SA cells are defective for ECAR increases (Figure 3A and Supp Figure 3). Yet they show that STAT3 SA cells have increased lactate production (Figure 3F). How is this possible, since lactate is what causes ECAR changes? Perhaps the answer relates to the fact that the increases in lactate production observed in STAT3 SA cells only approach the amount of lactate in the media of unstimulated Wild Type cells. In other words, the delta is not as important as the amount of lactate. Some discussion on this point would be helpful.

We would like to thank both reviewers for identifying this inconsistency as it raises an important issue regarding the contribution of lactate to extracellular acidification. As identified by both reviewers,

STAT3 SA mitochondria display reduced mitochondrial capacity due to the constitutive mutation of STAT3 Ser727 (Gough et al, 2009; Wegrzyn et al, 2009), highlighting the importance of STAT3 to mitochondrial function. Therefore, STAT3 SA mitochondria may be forcing glycolysis to maintain mitochondrial homeostasis and as such, the increased lactate concentration is reaching levels equivalent to wild type cells that is not indicative of increased lactate production. Reviewer 2 also raises the interesting alternative explanation of increased respiration and CO₂ to extracellular acidification, which was not the focus of this study, but may be the focus of future studies. We have discussed these issues within the text.

Page 12: Our data also show that STAT3 SA macrophages have lower basal ECAR, which is increased in response to LPS but not to the magnitude, observed in WT macrophages. This is in line with our previous observation that mitochondria from STAT3 SA cells are defective in ETC activity (Gough et al, 2009; Wegrzyn et al, 2009). Thus, STAT3 SA mitochondria may be more reliant on aerobic glycolysis that is consistent with the significant increase in the lactate concentration in STAT3 SA macrophages in response to LPS stimulation (Fig. 3g). However, the lactate concentration in LPS-treated STAT3 SA macrophages only ever approaches the lactate concentration observed in unchallenged WT macrophages. In addition, we do not observe any increase in the lactate concentration in WT macrophages in response to LPS stimulation. Given that ECAR is typically associated with lactate production this result appears somewhat counter-intuitive. However, it is important to note that ECAR also measures the export of CO₂, hydration of H₂CO₃ and the dissociation to HCO₃⁻ and H⁺ from the respiratory chain which also contributes to the ECAR reading (Mookerjee et al, 2105). Together, these data show that LPS induced STAT3 Ser727 phosphorylation is required for TCA cycle (succinate concentration) and OXPHOS (OCR) augmentation.

6. The data on IL-1 expression and protein abundance in macrophages (Figure 3) should be complemented by kinetic analyses and westerns. Why is IL-10 not an inducible protein in Figure 3J?

We have removed the data regarding IL-10 induction (formerly Fig 3j) and its accompanying TNF data (Fig 3i). To replace them we conducted kinetic analysis of IL-1 β , TNF α , IL-6 and IL-10 protein production in STAT3 SA peritoneal macrophages. This new data clearly enhances our understanding of the dynamic role STAT3 plays in mediating inflammatory cytokine production with distinct early regulation of IL-1 β . Interestingly, while IL-10 protein was increased from wild type macrophages, STAT3 SA macrophages were ablated in their upregulation of IL-10, consistent with STAT3 playing a role in regulating the induction of metabolically-dependent genes such as IL-1 β and IL-10. We describe these new results and their dynamic implications to inflammation in the text:

Page 13: We therefore, examined the kinetics and expression of cytokines in STAT3 SA PECs following LPS stimulation. Consistent with our mRNA data, while IL-1 β expression increased steadily between 4- 24 h of LPS challenge, IL-1 β protein expression was significantly suppressed in STAT3 SA cell lysates compared to WT cells (Fig 5b). Furthermore, TNF α expression was reduced in STAT3 SA compared to WT cells (Fig 5c), but did continue to increase parallel to WT expression, while IL-6 concentrations were only significantly different at 24 h post-LPS (Fig 5d). Interestingly, whilst we observed increased IL-10 expression in unstimulated STAT3 SA macrophages, they did not respond to LPS with the increase in IL-10 production observed in WT cells (Fig 5e).

Reviewer #2:

1) Based on the presence of binding motifs and IP assays, the authors conclude that TBK1 and STAT3 directly interact with TRAF6. This evidence is rather circumstantial. Additional experiments should be performed in which those binding motifs are mutated to see if the 'interaction' is lost.

We have now conducted mutational analysis of the STAT3-TRAF6 binding motif and identified residue E100 as critical to STAT3 interaction with TRAF6 (new Fig 1c)

2) *The data regarding a direct role for TBK1 in phosphorylating STAT3 are rather preliminary as they are based on inhibiting TBK1 using BX795, an inhibitor known to have some off-target effects. The inhibitor data should be validated in TBK1 silenced macrophages and in which TBK1 has a mutated TRAF6-binding motif.*

We agree with the reviewer that the use of BX-795 is not definitive in identifying TBK-1 as the functional kinase in STAT3 S727 phosphorylation. As requested, we obtained TBK-1 deficient macrophages and found that TBK-1-deficiency reduced STAT3 S727 phosphorylation by approximately 50%. However, combined with the BX-795 inhibitor data which significantly reduces pS727, these results suggest that IKK ϵ may also be involved. This would be consistent with the studies of Tan and Kagan (*Cell*, 2019) who conducted their studies in TBK-1 siRNA depleted IKK ϵ -deficient immortalized macrophages. Unfortunately we could not utilise immortalised IKK ϵ /TBK-1 double deficient macrophages as immortalized macrophages display constitutive STAT3 pS727 (please see explanation to Reviewer #1, Question 1 above) complicating clarity in ablating LPS-induced STAT3 pS727 and limiting our capacity to address this issue in the available time.

Pearce and colleagues also found that both IKK ϵ and TBK-1 were critical to LPS-induced glycolytic reprogramming in dendritic cells (*Everts et al, Nat Immunol, 2014*). At the current time we are unable to directly address this issue of TBK-1/IKK ϵ redundancy in STAT3 S727 phosphorylation, studies that we wish to address in the future. We discuss this possibility and the potential role of other kinases in the text (page 8 and 9 highlighted). We have also altered the title of the manuscript to acknowledging that TBK-1 may not be the primary kinase in STAT3 phosphorylation. We hope this satisfies the reviewer in the current circumstances.

3) *In general, WB signals should be quantified (phospho signal over total protein signal) as in several blots, the differences in signal are quite subtle.*

As suggested, we have now included densitometric analysis of the immunoblots to figures 2 and 3 to add clarity to our data and conclusions.

4) *In figure 1f, VDAC signal should be shown in the cytosolic fraction and tubulin in the mitochondrial fraction. Now it is unclear whether the fractions are pure or not.*

As suggested by Reviewer #1, we have removed these figures due to the quality of the fractionation and immunoblots. We have now replaced this figure with an immunoblot of mitochondrial fractions purified by centrifugation and included relevant fractionation immunoblots (new Fig 2f). We note that we had to conduct these experiments in immortalized BMDMs which resulted in increased background STAT3 pSer727 staining, however there is a clear increase in mitochondrial enrichment.

5) *TBK1 has been shown to promote early glycolytic programming through Akt-driven HKII activation in dendritic cells (Everts et al, 2014). Is this a route parallel to STAT3 through which TBK1 regulates metabolic reprogramming? What is the relative contribution? This is currently unclear. This should be tested, for instance are there additional metabolic effects TBK1 is inhibited/silenced in STAT3 SA cells?*

We have added additional discussion to the manuscript addressing the potential role and functional interaction of other kinases such as Akt and PI3 kinase in these pathways. It is currently unclear if the role of Akt phosphorylation of HKII and mitochondrial functionality operates parallel to STAT3 or is unique to dendritic cells. Our study focuses on identifying STAT3 as a novel mediator of LPS-induced macrophage metabolic reprogramming, potentially providing a mechanism to the observations of Kagan and colleagues describing a non-canonical role for TBK-1 in LPS-induced glycolysis via interaction with TRAF6. We agree that dissecting the contribution, cell specificity and kinetics of these signaling pathways moderating TLR-induced glycolysis is critically important. We have added

discussion related to these points in the manuscript and suggest these studies will form the basis of ongoing research. We believe that these aspects are beyond the scope of this current study.

Page 8-9: While PI3 kinase and Akt have previously been implicated in TLR-induced glycolytic reprogramming via TBK-1/IKK ϵ (Everts *et al*, 2014), none of the seven Akt inhibitors screened in our study reduced LPS-induced mtROS (Fig 3a). We therefore examined whether the PI3 kinase inhibitor TG100713 could also inhibit LPS-induced STAT3 Ser727 phosphorylation and noted that this inhibitor had no effect upon LPS-induced phosphorylation (Supplemental Data 1b). Therefore, while Akt has previously been identified as a downstream target of TBK-1/IKK ϵ in TLR-induced glycolysis in dendritic cells (Everts *et al*, 2014), neither Akt nor PI3 kinase were implicated in TBK-1-mediated glycolysis in macrophages (Tan and Kagan, 2019) and do not play a role in STAT3 S727 phosphorylation in macrophages. Together however, these studies identify a TRAF6, TBK-1/STAT3 signalling nexus leading to STAT3 S727 phosphorylation and mitochondrial localisation, providing a potential mechanism for the previously described role of TBK-1/IKK ϵ signalling in TLR-induced glycolysis (Everts *et al*, 2014; Tan and Kagan, 2019).

Indeed, while non-canonical TBK-1 recruitment and phosphorylation was identified as critical in macrophage LPS-induced glycolysis (Tan and Kagan, 2019), the role of Akt in macrophage signalling was not established akin to dendritic cells (Everts *et al*, 2014). Moreover, the role of these kinases may reflect cell specific signalling differences between macrophages and dendritic cells or reflect the kinetics of early and late glycolytic metabolic reprogramming pathways.

6) The authors state the 'data clearly show that S727 phosphorylation of STAT3 is critical to LPS-induced metabolic programming'. However, baseline mitochondrial metabolism is already significantly impaired in STAT3 SA cells. This makes it impossible to conclude that STAT3 is required for LPS-driven mitochondrial reprogramming, as these STAT3 SA cells seem to already have 'crippled' mitochondria.

It is true that STAT3 SA mitochondria are functionally decreased making it difficult to derive definitive conclusions from the data. However, it is not currently possible to induce the Ser to Ala mutation prior to LPS challenge *ex vivo* or *in vivo*. STAT3 SA mice present a viable model to explore the functional consequences of phosphorylation as compared to global gene deficiency or inhibitors that may have off target effects at higher concentrations. We have endeavoured to employ parallel experimental approaches such as identifying mediating kinases and interacting partners to provide an overview of the role of STAT3 in TLR-induced glycolysis. We discuss STAT3 SA mitochondrial limitations in the revised manuscript that are related to addressing Question 7.

7) In Fig S3 the authors find that STAT3 SA cells have impaired baseline and LPS-driven ECAR. It remains unclear how STAT3 would regulate ECAR (glycolysis). Or is in this case the difference in ECAR a reflection of reduced mitochondrial activity (hence CO₂ driven acidification)?

Both reviews identified this inconsistency as it raises an important issue regarding the contribution of lactate to extracellular acidification. As already identified by reviewer #2 (Q6 above), STAT3 SA mitochondria display reduced mitochondrial capacity due to the constitutive mutation of STAT3 Ser727 (Gough *et al*, 2009; Wegrzyn *et al*, 2009), highlighting the importance of STAT3 to mitochondrial function. Therefore, STAT3 SA mitochondria may be forcing glycolysis to maintain mitochondrial homeostasis and as such, the increased lactate concentration is reaching levels equivalent to wild type cells that is not indicative of increased lactate production. The suggestion raising the interesting alternative explanation of increased respiration and CO₂ to extracellular acidification, which was not the focus of this study, would be very interesting to explore in future studies. We have discussed these issues within the text.

Page 12: Our data also show that STAT3 SA macrophages have lower basal ECAR, which is increased in response to LPS but not to the magnitude, observed in WT macrophages. This is in line with our previous observation that mitochondria from STAT3 SA cells are defective in ETC activity (Gough *et al*, 2009; Wegrzyn *et al*, 2009). Thus, STAT3 SA mitochondria may be more reliant on

aerobic glycolysis that is consistent with the significant increase in the lactate concentration in STAT3 SA macrophages in response to LPS stimulation (Fig. 3g). However, the lactate concentration in LPS-treated STAT3 SA macrophages only ever approaches the lactate concentration observed in unchallenged WT macrophages. In addition, we do not observe any increase in the lactate concentration in WT macrophages in response to LPS stimulation. Given that ECAR is typically associated with lactate production this result appears somewhat counter-intuitive. However, it is important to note that ECAR also measures the export of CO₂, hydration of H₂CO₃ and the dissociation to HCO₃⁻ and H⁺ from the respiratory chain which also contributes to the ECAR reading (Mookerjee et al, 2105). Together, these data show that LPS induced STAT3 Ser727 phosphorylation is required for TCA cycle (succinate concentration) and OXPHOS (OCR) augmentation.

8) *The authors use HIF1a mRNA as a readout for inflammatory responses down stream of mitoROS. The papers they refer to show that ROS promotes HIF1a protein stabilization, not mRNA increase. The authors should assess HIF1a protein.*

Unfortunately, we were unable to assess HIF-1 α protein in STAT3 SA macrophages to support our mRNA data. We did not have enough STAT3 SA mice to produce sufficient peritoneal macrophages to conduct an immunoblot in addition to those required for other experiments due to a breeding phenotype inherent to STAT3 SA mice. We did endeavour to attempt HIF-1 α analysis in STAT3 SA macrophages by ELISA and immunostaining but were unsuccessful.

We have therefore removed the HIF-1 α mRNA data from the revised manuscript and removed any references to these results.

9) *Can the authors speculate how STAT3 is recruited to TRAF6? It already seems to be present in the traf6 complex (Fig 1a) before it is phosphorylated (Fig1b).*

Addressing the kinetics of the TRAF6/STAT3/TBK-1 signaling nexus, we see robust interaction between TRAF6 and STAT3 within 10 mins of LPS challenge (Lane 2, former Fig 1a, now Fig 1b), but minimal interaction in unstimulated cells (Lane 1). Consequently, we observe interaction between TRAF6 and TBK-1 within 10 mins of LPS stimulation (Top panel, Lane 2, Fig 3b). Subsequently, STAT3 interacts with TBK-1 after approximately 20 mins LPS stimulation (Top panel, lane 3, Fig 3c) which roughly corresponds with distinctive LPS-induced STAT3 S727 phosphorylation (Fig 2b).

We propose therefore that TRAF6 initially recruits STAT3 to the complex, prior to TBK-1 recruitment and subsequent phosphorylation of STAT3. Interestingly, Tan and Kagan (2019) also demonstrated TRAF6-dependent TBK-1 recruitment to the Mydososome within 15 mins post-LPS stimulation, consistent with the model proposed herein.

10) *The colorcoding of Fig 2a is not clearly indicated.*

We have added the appropriate color coding to the figure (now Fig 3a).

11) *In Figure 3 the authors show that LPS increases OCR. Yet, there are numerous studies showing that LPS leads to a reduction in mitochondrial respiration in myeloid cells over time. How can this be explained?*

We have now included text in the manuscript to address this (page 11):

Consistent with what we have previously observed in cancer cells (Gough et al, 2009), the loss of STAT3 S727 phosphorylation leads to a diminished basal respiration rate. Moreover, STAT3 SA macrophages had a significant reduction in their maximal respiratory capacity compared to WT cells (Fig. 4d-e). These data mean whilst STAT3 SA PECs are viable and actively respiring, they are

operating at their maximal capacity even in the absence of LPS stimulation. It should also be noted that we observe an increase in basal respiration following a 24 h LPS-treatment of peritoneal macrophages which is the opposite of the LPS-mediated suppression of OCR observed in BMDMs, but is consistent with other studies on PECs (*Artyomov et al, 2016; Rodriguez et al, 2019*). This potentially reflects the different polarisation of these macrophage populations, where PECs display higher expression of M1 markers when compared to BMDMs (*Bisgard et al, 2016*). Indeed, in addition to the intrinsic metabolic differences in macrophages from diverse microenvironments, it has been suggested that the process of culturing BMDMs for 7 days in M-CSF culturing may push them towards M2 differentiation as compared to peritoneal macrophages (*Wang et al, 2013*).

REVIEWERS' COMMENTS:

Reviewer #1 (Remarks to the Author):

In this revised manuscript, the authors have addressed all my concerns. I have no additional comments to offer.

Reviewer #2 (Remarks to the Author):

The authors have performed several new experiments and have added or rewritten several sections of the manuscript to address my comments. While not all concerns I raised in which I requested experimental validation, were actually addressed experimentally, the authors did tone down certain conclusions, have attempted to explain inconsistencies or pointed out limitations in experimental setup. This together with the new data and analyses, has resulted in a better balanced manuscript of overall higher quality, that in my view would now be suitable for publication in Nat Comm.

I have no further comments except for two minor textual points:

line 101: 'cells that was endemic to': this can be left out

Line 123: 'in' should be 'is'

Reviewer #1:

Thank you

Reviewer #2:

Thank you for assisting in improving the manuscript to provide a high quality and balanced study.

Minor textual points:

Line 101: 'cells that was endemic to' Has been removed

Line 123: 'in' has been corrected to 'is'.

Yours truly,

A/Prof Ashley Mansell, on behalf of Dr Daniel Gough

26/06/2020

Laboratory Head, Centre for Innate Immunity and Infectious Diseases

ashley.mansell@hudson.org.au

Ph. 0425 792 644